# The impact of environmental education at Chinese Universities on college students' environmental attitudes

Yibo Li *, Dongli Yang, Siyuan Liu

School of Marxism, China University of Geosciences (Beijing), Haidian District, Beijing, P.R. China

* 3018200005@email.cugb.edu.cn

**Data Availability Statement:** Data Fetch URL: https://github.com/LeeYibo/Dissertation-data-on-ecological-attitudes.

**Funding:** The author(s) received no specific funding for this work.

## Abstract

The purpose of this study is to examine the effects of environmental education on students' attitudes about the environment in Chinese higher education. The findings showed that students' environmental attitudes can be greatly enhanced by college-level ecology and environmental education. One of the most major factors influencing students' environmental attitudes in the context of college environmental education is subjective norms, and curriculum education also has a big impact on this. It is possible that Chinese college students today lack the self-efficacy necessary to safeguard the environment since perceived behavioral control has less of an impact on college students' environmental attitudes than subjective norms and curricular education. This highlights the need of promoting environmental practices and improving college students' self-perceive and capacity for environmental protection. The study also showed that factors including gender, location, educational level, and economic status of the family had no impact on college students' environmental attitudes. The results of this study can be used to examine the factors influencing the environmental views of Chinese college students and to teach educators how to raise college students' awareness of the environment through curricular modifications, classroom instruction, and perceived behavioral control.

## 1. Introduction

The Chinese government has increased its emphasis on the significance of environmental quality as a key component of sustainable development as blind, excessive, and disorderly resource exploitation and unrestrained consumption of resources gradually approach and even exceed the limits of environmental carrying capacity [1]. By raising public awareness, the government hopes to achieve the goal of sustainable development, where the economy, resources, and ecological environment are coordinated. However, environmental attitudes and behaviors are at a low level in Chinese society [2].

According to relevant study findings, education is a key strategy for achieving sustainable development and raising citizen quality [3]. The National Bureau of Statistics of China reports that throughout the previous five years, the number of general undergraduate, master's, and

**Competing interests:** The authors have declared that no competing interests exist.

doctoral students in China has increased. China will have 36.594 million general undergraduate and specialized students as of 2022, compared to 3.654 million postgraduate students (including master's and PhD students) [4]. According to the aforementioned data, college students have become a significant demographic in Chinese culture, and their ecologically literate attitudes have a significant impact on how green and healthy the country as a whole is [5]. The People's Republic of China's Ministry of Education (Ministry of Education, PRC), which oversees education, has released several policy documents and standards related to green education and environmental education. As a result, many colleges and universities have promoted college students' awareness and attitudes of valuing the environment, being close to the environment, and protecting the environment through ecological and environmental education.

An essential objective of green education at colleges and universities is to foster students' cognitive attitudes toward the environment [6]. This cognitive attitude has a direct impact on how they live their lives. Over time, there have been continuous efforts to enhance ecological education in institutions of higher education, aiming to inculcate proper ecological ethics and morals in college students. The results of existing studies widely recognize the role of green education in higher education institutions in shaping the cognitive attitudes of college students [7]. According to a study by McMillan et al., students' environmental values dramatically increased after taking a course on environmental conservation, leading to more favorable cognitive attitudes [8]. This demonstrates the critical role that environmental education plays in helping students develop their worldview, ethics, and values, which can boost their motivation and willingness to engage in both individual and group environmental actions [9].

According to the Theory of Planned Behavior (TPB) put forth by Ajzen et al., a person's attitude toward a particular behavior or problem is influenced by perceived behavioral control and subjective norms [10]. Subjective norms refer to the pressure or expectation that is exerted on an individual by the perceptions of other people in the individual's social circle towards a particular behavior [11], and perceived behavioral control refers to an individual's perception of the ease or difficulty in carrying out a particular behavior, if a person perceives a behavior to be easy and within their ability, then they are more likely to have a positive attitude towards that behavior. Conversely, if individuals believe the action to be challenging or beyond of their control, they are more likely to view it negatively [11]. While offering curricular instruction, higher education also creates a more cohesive social network for college students, and this social circle will affect how they feel about the surroundings [12]. Therefore, when analyzing the influence of higher education on college students' environmental attitudes, subjective norms and perceived behavioral control variables should also be taken into consideration when examining the impact of higher education on college students' environmental attitudes. The findings of previous studies demonstrate that education, subjective norms, and perceived behavioral control all significantly influence people's attitudes toward their environment. It is important to think about and explore how these social and behavioral control factors interact to affect college students' attitudes toward the environment. This study will also examine the connection between a few demographic traits and the environmental attitudes of college students.

This study identifies the following research questions considering the research hypotheses and findings:

1. Are students' environmental attitudes affected by demographic factors?

2. Does school curriculum education affect students' environmental attitudes?

3. Do subjective norms in the schooling process affect students' environmental attitudes?

4. Does perceived behavioral control in the schooling process affect students' environmental attitudes?

## 2. Literature review

### 2.1. Research on the relationship between environmental education and personal environmental attitudes

The findings of numerous studies have demonstrated that individual age [13], educational level [3, 13], geographic location [14], cultural environment [15], and family education [16] are all common influences on people's attitudes and intentions toward the environment. Schooling is typically seen as being more important than the other factors [17]. In order to establish the concept of green environmental protection at the subjective level and be willing to adopt behaviors to protect the environment in their daily lives, people can establish a clear understanding of ecological knowledge and the significance of protecting the environment through education.

The transmission of environmental knowledge through education is a key channel for influencing students' environmental attitudes [2], according to research on environmental protection education, which demonstrates that targeted education in schools can effectively influence students' attitudes and behaviors toward the environment [18]. According to a study by McMillan et al., there is a significant positive correlation between academic achievement and students' attitudes toward the environment. The study found that students' environmental values and attitudes toward environmental protection significantly improved after completing an environmental protection program [8].

Based on the Theory of Reasoned Action and the Theory of Planned Behavior, Suárez-Perales et al. examined the influence of higher education on pro-environmental attitudes and behaviors. It was discovered that the knowledge-concern-will model is a good representation of the impact of education on students' environmental attitudes [3]. In other words, it is by learning about environmental protection that kids become aware of the environment, develop a desire to protect the environment, and develop behaviors that support this desire. Based on the idea of training needs, Lei et al. examined the elements influencing people's willingness to live sustainably. The findings revealed that education and training can increase people's environmental knowledge and increase their willingness to live sustainably [19]. Environmental education can help people develop positive environmental attitudes by enabling them to gain scientific understanding about environmental protection, raising the significance they place on environmentally friendly behavior, and recognize the advantages of doing so. Tai-Yi Yu and Tai-Kuei Yu et al. constructed an assessment model to analyze the significant effects of sustainability values, environmental issues, social norms, perceived risks, and pro-environmental attitudes on students' participation in environmental behavior [20].

Teaching strategies have a significant role in determining how effective education is. Claudia Borchers et al. shown how encouraging student involvement in the extra-curricular environmental education can significantly increase students' pro-environmental willingness [21]. Based on Bogner and Wiseman's Model of Ecological Values, Bruce Johnson discovered that encouraging students to engage in practical ecological protection-related activities can significantly improve their views of and attitudes about environmental preservation [22]. According to Zhang Hongxia and colleagues, enhancing the design of related courses can enhance the effectiveness of peer education within the student body, create an educational environment that is conducive to the construction of an ecological civilization, encourage college students to adopt healthy lifestyles, and ultimately help them to develop favorable attitudes and perceptions of the environment [23].

The aforementioned study's findings demonstrate that the influence of environmental education on students' environmental attitudes has been extensively acknowledged [7]. However, fewer studies on the connection between higher education and college students'

environmental attitudes than there are on the research and analysis for research and analysis for primary and secondary school students. Although the years spent in elementary and secondary school are crucial for the development of personal values, the reshaping of values throughout the higher education stage should not be disregarded. Based on the existing research results, we propose a hypothesis to examine the influence of university education on college students' environmental attitudes.

Hypothesis 1 (H1): Attitudes toward the natural environment among college students will be influenced by school curriculum education.

## 2.2. Studies on the relationship between subjective norms and personal environmental attitudes

The Theory of Planned Behavior (TPB) suggests that there is a significant relationship between subjective norms and individual attitudes and behaviors [10], and that both imperative and descriptive norms have a significant impact on individual intentions and attitudes [11, 24]. By analyzing a sample from Jordan, Alshurideh et al. discovered that subjective norms significantly influence a person's attitudes and intentions to engage in a particular action [25]. This conclusion is consistent with that of Kara, who using the Theory of Planned Behavior, examined how the government of Hong Kong, China, encouraged waste recycling and found that attitudes are the primary predictor of whether or not an individual will adopt a particular behavior, and that subjective norms have a significant impact on an individual's attitudes [26]. Social pressure-induced subjective norms had a greater effect on people's environmental attitudes and behaviors, especially after the COVID-19 pandemic [27]. Although the findings of Naz Onel's study on environmental consumption in the United States indicated that personal norms have a stronger influence on individual attitudes and behaviors than subjective norms, the study's findings also supported the impact of subjective norms on people's intentions to engage in particular behaviors [28].

By examining the development of children's pro-environmental behaviors in various cultural contexts in Germany and Japan, Kaori Ando et al. discovered that parents play a significant role for children. Therefore, parents' subjective norms have a significant impact on their children's pro-environmental attitudes [29]. Children are more vulnerable to subjective norms than adults, according to their study's findings. In other words, psychologically immature groups are more significantly impacted by subjective norms than are adults.

According to psychosocial development theory, college students are neither children nor fully grown adults; rather, they are in a stage of transition from adolescence to adulthood [30]. During this time, college students' cognition and values are not stabilise, and they are easily impacted by a variety of social influences, most notably the expectations that society has of them [31]. To examine the connection between subjective norms and the environmental attitudes of college students, we therefore present a hypothesis.

Hypothesis 2 (H2): Attitudes toward the natural environment among college students will be influenced by subjective norms in the educational process.

## 2.3. Research on the relationship between personal environmental attitudes and perceived behavioral control

Perceived behavioral control is frequently thought to play a significant role in attitudes, which include things like the skills and abilities required to perform a specific behavior. Externally provided strong attitudes and supportive norms can also help to form an individual's intention

to engage in a particular behavior [11]. Attitudes toward a behavior are the expression of a person's positive or negative attitude toward the performance of a specific behavior. According to Godin et al., perceived behavioral control factor, on the other hand, reflects how easy or difficult it is for a person to perform this behavior and reflects the influence of external and internal factors on an individual's intentions and attitudes [32]. By creating a theoretical model of planned behavior, Saeid Karimi et al. conducted a study on students' environmental intentions and behaviors in a developing nation (Iran) and discovered that perceived behavioral control has a significant and positive correlation with the students' environmental intentions and behaviors. Additionally, this has the greatest influence [33]. This conclusion was further supported by a study from Sweden [34].

However, out of all the elements that affect people's behavior, perceived behavioral control does not always have the greatest influence. Social norms have been demonstrated in studies to have the greatest influence on farmers' environmental behavior, with perceived behavioral control having a secondary effect [35]. By comparing and contrasting the factors influencing people's environmental behavior in various cultures, Kaori Ando et al. analyzed the factors influencing people's environmental behavior in various cultures and discovered that, in contrast to Japan, which places a premium on interpersonal relationships, perceived behavioral control in the data sample from Germany revealed a more significant role [36].

According to the results, perceived behavioral control exerts varying degrees of influence on various populations. But in any case, perceived behavioral control is an important factor influencing people's behavioral attitudes. Therefore, we propose the following hypotheses:

Hypothesis 3 (H3): Attitudes toward the natural environment among college students will be influenced by perceived behavioral control in the educational process.

## 3. Materials and methods

### 3.1. Measures and measurement

Based on the research subject, which examined the impact of environmental education at Chinese universities on college students' environmental a ttitudes, the topics included in the scales for this study were derived from earlier research [37–41] and were suitably changed to meet the specific context of the study. Environmental attitudes (EA), subjective norms (SN), perceived behavioral control (PBC), and school curriculum education (SCE) were the four dimensions of the study. Respondents indicated their degree of agreement with the statements on a scale from 1 (strongly disagree) to 5 (strongly agree) for each of the questionnaire's five-point Likert scale items. In this study, the questionnaire was administered to the respondents through an online format and the respondents were prompted to give informed consent before completing the research questionnaire; therefore, the form of obtaining consent was electronic. No minors participated in the study.

### 3.2. Data sources and basic information about the sample

The formal research for this study was conducted from 24th and 29th May 2023. The participants in the study were first informed of its goals and the nature of the questionnaire. The pupils then willingly participated and answered the questionnaire, sharing their perspectives on each dimension. Before the official research began, 50 participants completed a pre-survey. Based on the findings of the pre-survey and respondent interviews, the final questionnaire was changed. There were 400 responses to the formal survey. A total of 398 valid replies were kept for analysis after incorrect data was removed. The distribution of participants, which fit the

fundamental criteria of the study's target demographic, was equal based on gender, educational level, and hometown area (Tabel 1).

### 3.3. Reliability testing of the questionnaire

Scales were used in this study to measure the key variables. Therefore, evaluating the measurement's data quality was essential to verify that the analysis that followed. The Cronbach's alpha (α) coefficient reliability test was used to assess each dimension's internal consistency [42]. Cronbach's alpha (α) coefficients range from 0 to 1, and higher values indicate that the questionnaire is more reliable and consistent. Reliability coefficients below 0.6 are typically seen as untrustworthy, while coefficients between 0.6 and 0.7 are thought to be credible, 0.7 and 0.8 are thought to be somewhat credible, 0.8 and 0.9 are thought to be more credible, and 0.9 and 1 are thought to be extremely credible.

The Cronbach's alpha (α) coefficient, a measure of internal reliability, was examined using SPSS software. The data used for the analysis are shown in the data file in the S1 File. The findings revealed that the EA, SN, PBC, and SCE dimensions' respective Cronbach's alpha (α) coefficients were 0.756 ($>$0.7), 0.679 ($>$0.6), 0.722 ($>$0.7), and 0.776 ($>$0.7). Additionally, the questionnaire's total Cronbach's Alpha (α) value was 0.914 ($>$0.9). These findings suggest that the examined variables had strong internal consistency and dependability (Table 2).

## 4. Results

The results of the survey show that 53.8% of the male university students interviewed in this research and 46.2% of the female university students were interviewed. In terms of educational level, 45.7% of the respondents were specialized students, 46.7% were undergraduates, and 7.5% were master's or doctoral students. In terms of hometown location, 54.8% of the respondents were from rural areas and 45.2% were from cities. Meanwhile, the level of average monthly living expenses shows to a certain extent the financial status of the respondents' families. 22.9% of the interviewed students are under Rmb1,500, 51.0% are between Rmb1,500–2,000, 18.8% are between Rmb 2,001–3,000, and 7.3% are above Rmb 3,000 (Table 1). This study was analyzed using SPSS statistical analysis software to examine the status of college students' environmental attitudes and the status of the school's ecological and environmental education development, and to measure the effects of demographic factors, school curricular education, subjective norms, and perceived behavioral control factors on students' perceived attitudes toward environmental sustainability.

**Table 1. Sample basic information.**

| | Valid | Frequency | Percent (%) |
|---|---|---|---|
| Gender | Male | 214 | 53.8 |
| | Female | 184 | 46.2 |
| Educational level | Specialties | 182 | 45.7 |
| | Undergraduate | 186 | 46.7 |
| | Master's degree and above | 30 | 7.5 |
| Hometown location | Rural areas | 218 | 54.8 |
| | Urban areas | 180 | 45.2 |
| Monthly living expenses (Rmb) | Up to 1,500 (excl.) | 91 | 22.9 |
| | 1,500–2,000 | 203 | 51.0 |
| | 2,001–3,000 (excl.) | 75 | 18.8 |
| | 3,000 or above | 29 | 7.3 |

**Table 2. Scale variables, questions, and internal reliability.**

| Variable | Item | Cronbach's alpha (α) |
|---|---|---|
| EA | Selecting eco-friendly and low-carbon behaviors in daily life holds significance. | 0.756 |
| | Embracing eco-friendly and low-carbon behaviors in daily life embodies a positive lifestyle. | |
| | Being willing to compromise to safeguard the environment. | |
| | Efforts should be made to conserve resources and protect the environment to the best of one's ability. | |
| SN | Peers and classmates around you believe that opting for eco-friendly and low-carbon behaviors is advisable. | 0.679 |
| | Your classmates, teachers, and family members frequently choose eco-friendly and low-carbon behaviors. | |
| | Your college actively encourages the adoption of eco-friendly and low-carbon behaviors in your daily life. | |
| PBC | It is convenient for you to opt for eco-friendly behaviors. | 0.722 |
| | The decision to choose eco-friendly behaviors is within your discretion. | |
| | You are inclined to choose eco-friendly behaviors if it doesn't lead to losing face in front of those around you. | |
| | You possess the knowledge and capability to practice eco-friendly and low-carbon behaviors. | |
| SCE | Colleges offer specialized courses on ecological and environmental conservation education. | 0.776 |
| | The college provides comprehensive and in-depth knowledge of ecology and environmental protection. | |
| | Colleges frequently organize initiatives to preserve resources and safeguard the environment. | |
| | Colleges have displayed slogans promoting resource conservation and environmental protection. | |
| | Colleges frequently host lectures on resource conservation and environmental protection. | |

## 4.1. The environmental attitudes of college students

The sample's opinions regarding environmentally friendly and low-carbon actions are revealed by the environmental attitudes (EA) dimension questions. The findings reveal that the research sample's average environmental attitude score is 4.40, indicating that university students have highly favorable attitudes about engaging in environmentally friendly practices. The findings (Table 3) demonstrate that Chinese college students have a supporting attitude toward environmental protection by looking at the frequency of the four items contained in the pro-environmental attitude dimension. More than 90% of college students think it is important to live a green and low-carbon lifestyle. Nearly 88% of college students agree that leading a green or low-carbon lifestyle is a good idea and are eager to do their part to conserve resources and safeguard the environment. Approximately 85% of college students claimed they are willing to make sacrifices in their daily life to safeguard the environment, such as:

**Table 3. Frequency (%) distribution of college students' environmental attitudes.**

| | Strongly Disagree | Disagree | Moderately | Agree | Strongly Agree |
|---|---|---|---|---|---|
| Selecting eco-friendly and low-carbon behaviors in daily life holds significance. | 0.3 | 1.8 | 6.0 | 28.9 | 63.1 |
| Embracing eco-friendly and low-carbon behaviors in daily life embodies a positive lifestyle. | 0 | 2.0 | 9.8 | 38.7 | 49.5 |
| Being willing to compromise to safeguard the environment. | 0 | 1.0 | 13.6 | 36.2 | 49.2 |
| Efforts should be made to conserve resources and protect the environment to the best of one's ability. | 0.3 | 1.3 | 10.3 | 35.2 | 53.0 |

**Table 4. The influence of demographic factors on college students' environmental attitudes.**

| | Sex | | educational level | | | Hometown Location | | Monthly living expenses(Rmb) | | | |
|---|---|---|---|---|---|---|---|---|---|---|---|
| | Male Mean | Female Mean | Specialties | Under graduate | Master's degree and above | Rural areas | Urban areas | Up to 1,500 (excl.) | 1,500 - 2,000 | 2,001 - 3,000 (excl.) | 3,000 or above |
| EA | 4.45 | 4.35 | 4.41 | 4.38 | 4.54 | 4.42 | 4.39 | 4.29 | 4.42 | 4.44 | 4.47 |
| Std. Deviation | 0.58 | 0.53 | 0.50 | 0.62 | 0.53 | 0.59 | 0.53 | 0.69 | 0.52 | 0.47 | 0.53 |
| Total Mean | 4.40 | | 4.40 | | | 4.40 | | 4.40 | | | |
| T/F | 1.95 | | 1.17 | | | 0.53 | | 1.59 | | | |
| Sig. | 0.052 | | 0.310 | | | 0.596 | | 0.191 | | | |

traveling as often as possible on public transportation, avoiding the use of throwaway things, conserving water, and electricity resources, and so on. There are also some university students who have a negative attitude towards environmental protection. 2.0% of college students disagree with green and low-carbon lifestyles and activities, and they do not believe that they should make every effort to conserve resources and support environmental protection. 1.0% of college students disagree with the practice of giving in to environmental protection.

## 4.2 The impact of demographic variables on the environmental attitudes of college students

One-way ANOVA was used to examine the effects of demographic variables on college students' pro-environmental attitudes. This analysis looked at how gender, educational attainment, hometown location, and average monthly cost of living (family income status) affected these attitudes. According to the findings (Table 4), gender, educational attainment, hometown location, and average monthly living expenses had no statistically significant impact on college students' environmental opinions.

## 4.3. Ecological and environmental education in universities and colleges

The mean values of the three aspects of the subjective norms (SN) perceived behavioral control (PBC) school curriculum education (SCE) were used to evaluate and examine the state of eco-environmental education in China. According to the findings (Table 5), college students' mean scores across the three dimensions of subjective norms, perceived behavioral control, and curriculum and education were, respectively, 4.37, 4.26, and 4.36. The mean value of the subjective norms is 4.37, indicating that college students' social environment, as well as the cognitive attitudes and requirements of those around them, have a significant influence on their environmental attitudes. In other words, the environmental attitudes and practices of college students' friends, teachers, and other individuals around them have an impact on their environmental attitudes. In addition, the perceived behavioral control factor's mean value is 4.26, suggesting that few college students will give up their environmental habits out of embarrassment or concern for their reputation. It should be recognized, nevertheless, that not

**Table 5. Subjective norms, perceived behavioral control, and curriculum education scores.**

| | N | Minimum | Maximum | Mean | Std. Deviation |
|---|---|---|---|---|---|
| SN | 398 | 2.00 | 5.00 | 4.37 | 0.57 |
| PBC | 398 | 2.00 | 5.00 | 4.26 | 0.58 |
| SCE | 398 | 1.80 | 5.00 | 4.36 | 0.56 |

**Table 6. Relevance analysis.**

|  | EA | SN | PBC | CE |
|---|---|---|---|---|
| EA | 1 |  |  |  |
| SN | 0.730** | 1 |  |  |
| PBC | 0.648** | 0.677** | 1 |  |
| SCE | 0.727** | 0.732** | 0.713** | 1 |

Note
**. P<0.01 (two-tailed).

everyone will always act in a way that is consistent with their claimed objectives. Although respondents claimed that they were free to make their own decisions and that no outside pressure or praise would influence their opinions on environmental concerns, actual behaviors may differ [40]. As a result, the impact of perceived behavioral control on college students' environmental attitudes may differ from what is the case. The average score for education in the school curriculum was 4.36, which shows that, as a result of national development requirements and policies, environmental education is now more prevalent in China's higher education system.

## 4.4. Impact of higher education eco-environmental education on college students' environmental attitudes

The variables were compared using a preliminary correlation analysis, and the findings (Table 6) revealed a significant positive correlation between the four dimensions of college students' environmental attitudes and subjective norms, perceived behavioral control, and educational elements of the school curriculum (at the 0.01 level of significance). The individual environmental attitudes (EA) was found to be most influenced by subjective norms (SN), followed by school curriculum education (SCE) factors, and least influenced by perceived behavioral control (PBC) factors, according to Pearson's correlation coefficients between the variables and behaviors.

The current study used the methodology outlined by Cohen et al. [43], a nd performed linear regression analyses on the sample data in order to further investigate the precise influences of school curriculum education (SCE), subjective norms (SN), and perceived behavioral control (PBC) on college students' environmental attitudes (see Table 7). With environmental

**Table 7. Linear regression results.**

|  | Unstandardized Coefficients | | Standardisation Coefficients | t | P | VIF |
|---|---|---|---|---|---|---|
|  | B | Std. error | β |  |  |  |
| (Constant) | 0.710 | 0.147 |  | 4.832 | 0.000** |  |
| SN | 0.367 | 0.047 | 0.377 | 7.836 | 0.000** | 2.410 |
| PBC | 0.140 | 0.045 | 0.146 | 3.116 | 0.002* | 2.270 |
| SCE | 0.342 | 0.050 | 0.347 | 6.865 | 0.000** | 2.654 |
| $R^2$ | 0.621 | | | | | |
| F | 215.486 | | | | | |

Dependent variable: EA
Note
*. P<0.05
**. P<0.01

attitudes (EA) acting as the dependent variable, subjective norms (SN), perceived behavioral control (PBC), and school curriculum education (SCE) were used as independent variables. There is no issue of multicollinearity among the independent variables, according to the regression model with an $R^2$ value of 0.621, which also demonstrated a good fit, a F value of 215.486, and a VIF value of less than 5, indicating that each independent variable influences college students' attitudes toward the environment independently and that all of them have a positive impact. The model coefficients indicate that subjective norms (SN) followed by school curriculum education (SCE)have the greatest influence on college students' environmental attitudes (EA), followed by school curriculum education (SCE). These two factors were significantly correlated at the 0.01 level. The least impactful factor is perceived behavioral control (PBC), which is significantly correlated at the 0.05 level.

## 5. Discussion

The study's findings shed light on Chinese college students' views toward the environment as well as the variables that shape such opinions. The high degree of environmental sentiments among college students points to a rising trend in the population's environmental consciousness. This is driven in part by the richer ecological and environmental protection curricula offered in colleges and universities, which emphasizes the critical function of higher education in reinforcing environmental protection views among college students.

The results of the study show that the current Chinese college students have more positive environmental attitudes, which is consistent with the results of previous studies [44]. According to the study's findings, impacts of gender, hometown location, educational attitudes, and monthly living expenses on college students' environmental attitudes were not significant, which differed from the results of previous studies [1, 3, 13]. This conclusion suggests that the younger generation of university students are gradually moving beyond the conventional restrictions of gender and other factors in their concern for environmental issues as China's economy and society continue to develop and education levels continue to increase.

Overall, China's higher education has gradually created more comprehensive environmental education programs in order to support students in forming positive attitudes toward the environment. With correlation coefficients of 0.730, 0.727, and 0.648, which supported the findings of the earlier studies, the linear regression results of this study demonstrated that subjective norms (SN), school curricular education (SCE), and perceived behavioral control (PBC) in the educational process had a significant effect on the environmental attitudes (EA) [2, 5, 9, 45] of Chinese college students.

It was found that the most influential factor on college students' environmental attitudes (EA) in Chinese society was subjective norms (SN). This finding confirms previous research [33, 46, 47], and demonstrates that social pressure, peers' and teachers' environmental attitudes and behaviors, as well as expectations of college students' environmental attitudes and behaviors, have a significant impact on college students' environmental attitudes. As a result, the hypothesis that university students' attitudes towards the natural environment are influenced by subjective norms in the educational process (H2) is supported. This highlights the value of classmates, teachers, and other members of college students' social circles being aware of their environmental responsibilities in order to improve their environmental attitudes. Therefore, it is crucial to stress the improvement of peer education during the ecological and environmental education process at colleges and universities. Peer education, as a useful addition to traditional classroom and practical instruction, can free up teachers' time and attention to better meet the needs of their students [48], and can offer college students constructive subjective guidance to hasten their understanding of the fundamental ideas underlying green

environmental protection as well as the development of underlying cognitive attitudes through cooperation and communication among peers. Although peer education aids in the development of constructive cognitive attitudes and behavioral patterns, teachers must still be involved. Teachers' low-carbon environmental attitudes and practices have a favorable effect on students' cognitive attitudes and values because they are, in some cases, seen as important or respectable by college students. As a result of receiving relevant education, students not only develop more favorable cognitive attitudes and behavioral practices toward the ecosystem, but their thought processes also change, encouraging them to lead more sustainable lifestyles [8].

The study's findings also demonstrated the important influence of school curriculum education (SCE) on Chinese university students' Environmental attitudes(EA). This means that college students' attitudes toward environmental protection will be more favorable if schools provide specialized ecological and environmental protection education courses, teachers provide systematic and in-depth explanations of pertinent knowledge, frequently hold environmental protection lectures and practical activities, and establish a cultural atmosphere of environmental protection. The hypothesis that university students' attitudes towards the natural environment would be influenced by education in the school curriculum was also tested (H1). The promotion of environmental values and the growth of environmentally conscious attitudes and actions are both possible outcomes of environmental education in schools [49]. Therefore, it is imperative that universities integrate thorough instruction in ecological and environmental protection into their curricula by offering specialized environmental education courses led by specialized instructors, as well as by delivering thorough lectures and encouraging student participation. Implementing a thorough and in-depth education system can enhance university students' cognitive attitudes about environmental sustainability, as well as foster their sense of responsibility for the environment and their capacity to defend it [50]. A significant amount of research has also shown how multimedia technologies have a major impact on daily living, emphasizing their impacts on perceived behavioral control. People's interactions with movies and social media platforms, in particular, have a significant impact on how they perceive different events and make decisions [51]. Schools should include digital technologies, movies, and television shows into their ecological and environmental education curricula. These media-based resources can be used as supplemental instruction for college students, giving them the opportunity to relate the material to social practices currently practiced in Chinese universities, apply what they have learned in the classroom to real-life situations, gain a deeper understanding of environmental issues and knowledge, and improve their environmental attitudes.

In comparison to subjective norms (SN) and school curriculum education (SCE), the study's findings indicated that perceived behavioral control (PBC) had a lesser overall impact on college students' environmental views, but it did so in a statistically significant way ($p < 0.05$). This finding shows that although while perceived behavioral control has a minor impact on college students in contemporary Chinese society, it nevertheless has a statistically significant effect. In addition to behavioral feasibility, social cognition, personal values, and perceived self-efficacy also have an impact on college students' attitudes toward the environment. Environmental views among Chinese university students are influenced not just by behavioral feasibility, but also by social cognition, personal values, and perceived self-efficacy. The hypothesis (H3) that perceived behavioral control during the learning process influences college students' views toward the natural world is supported. The outcomes of this study also corroborate those of Kaori Ando et al. [36]. This situation suggests that university students in Chinese society have positive attitudes toward environmental protection, but that these attitudes are difficult to translate into actions due to the influence of limiting factors, which is also

consistent with the findings of earlier research [52]. Psychological, social, and cultural factors influence how people feel about the environment. When college students have doubts about their capacity to engage in environmentally friendly behaviors, they feel less in control of those behaviors, which hinders their capacity to translate their feelings about the environment into actual, practical actions. The power of perceived behavioral control is further diminished by subjective norms, which lead Chinese college students to consider these norms rather than their own attitudes while dealing with environmental protection issues. Although it should be highlighted that the influence of perceived behavioral control factors on pro-environmental attitudes is less in comparison to those of Curricular education and subjective norms, it is still important to notice. Individual attitudes and behaviors can be affected by perceived behavioral control, which has been well-documented [53].

The findings of this study therefore suggest that to close the gap between environmental attitudes and sustainable actions, the environmental education process in Chinese universities must improve college students' perceptions of control over their environmental behaviors. It is possible to increase students' perceptions of control over environmental protection by breaking down environmental goals into manageable activities to increase students' self-assurance, holding training sessions to provide students with the knowledge and skills to implement environmental behaviors, and motivating students to actively engage in environmental activities through rewards in the classroom to strengthen environmental attitudes and behaviors.

## 6. Conclusion

In conclusion, this study reveals that Chinese college students generally had Positive attitudes about environmental protection, and ttheir sentiments were not significantly impacted by demographic criteria including gender, education level, hometown location, and standard of living level. These circumstances imply that in the current Chinese society, environmental concerns are growing more common among college students as a group.

According to this study, The main Influencing factors of college students' environmental attitudes (EA) in contemporary Chinese higher education are subjective norms (SN) and school curriculum education (SCE). This emphasizes how important it is for the social environments and curriculum education on college students' attitudes toward environmental protection. It is for this reason suggested that during the environmental educational process, colleges and universities concentrate on developing students' intrinsic environmental values and encourage their environmental consciousness and sense of responsibility by providing them with guidance about subjective norms. The study also highlights how important perceived behavioral control (PBC) is in schools. To encourage students' interest and enthusiasm in learning about environmental protection and engaging in environmental protection activities, colleges and universities should work to enhance their students' perceptions of environmental protection and their self-perceived efficacy in the environmental education process. Students' environmental knowledge will increase as a result, and their practical environmental protection skills will grow.

Based on the results of the study, in order to promote the awareness and attitude of environmental protection among college students, it is advised that Chinese colleges and universities focus on the role that environmental education plays in influencing students' attitudes toward the environment and support the in-depth development of environmental education by integrating educational resources and fostering an environment on campus that motivates students to adopt environmentally friendly attitudes and behaviors. Overall, this study has good implications for the creation of more environmentally conscious college students and offers strong support for Chinese colleges and universities to create focused environmental education strategies.

## 7. Limitation and suggestions

In analyzing the effects of direct and indirect educational factors on college students' perceived green attitudes, this study introduced perceived behavioral control factors into the analytical framework, which effectively supplemented the factors influencing the effectiveness of green education in schools. However, the sample size of the study is relatively small compared to the large college student population in China, so the results can only reflect the situation within the surveyed region to a certain extent, and college students from specific geographic regions or socioeconomic backgrounds may be overrepresented, thus affecting the generalizability of the results to all college student groups in China. The study also focuses on the opinions and attitudes of the college students who were interviewed at a particular time for analysis, and this self-reporting may be influenced by psychological factors, i.e., participants may give answers that they believe to be socially acceptable rather than reflecting their true attitudes and behaviors. Additionally, people's opinions and behaviors change with time. To produce more focused and generalizable results, future studies must therefore increase the sample size or carry out targeted studies based on regions. To better understand the difficulties and chances for enhancing college students' environmental attitudes in China's multicultural context, it is also important to study the impact of cultural values on attitudes and behaviors.

## Supporting information

**S1 File. Data for analysis.**
(XLSX)

## Author Contributions

**Conceptualization:** Yibo Li, Dongli Yang.

**Data curation:** Dongli Yang, Siyuan Liu.

**Formal analysis:** Yibo Li.

**Investigation:** Yibo Li, Siyuan Liu.

**Methodology:** Yibo Li.

**Writing – original draft:** Yibo Li.

**Writing – review & editing:** Yibo Li.

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
