## [Decision Letter · Decision Letter 0]

14 Dec 2023

PONE-D-23-29584The Impact Environmental Education at Chinese Universities on College Students' Environmental AttitudesPLOS ONE

Dear Dr. Li,

Thank you for submitting your manuscript to PLOS ONE. After careful consideration, we feel that it has merit but does not fully meet PLOS ONE’s publication criteria as it currently stands. Therefore, we invite you to submit a revised version of the manuscript that addresses the points raised during the review process.

We look forward to receiving your revised manuscript.

Kind regards,

Bo Pu, Ph.D.

Academic Editor

PLOS ONE

Journal Requirements:

3. During your revisions, please note that a simple title correction is required: Your title needs to be changed to "The Impact of Environmental Education at Chinese Universities on College Students' Environmental Attitudes". Please ensure this is updated in the manuscript file and the online submission information.

Additional Editor Comments (if provided):

Reviewers' comments:

Reviewer's Responses to Questions

**Comments to the Author**

1. Is the manuscript technically sound, and do the data support the conclusions?

Reviewer #1: Partly

Reviewer #2: Yes

2. Has the statistical analysis been performed appropriately and rigorously? 

Reviewer #1: Yes

Reviewer #2: Yes

3. Have the authors made all data underlying the findings in their manuscript fully available?

Reviewer #1: Yes

Reviewer #2: Yes

4. Is the manuscript presented in an intelligible fashion and written in standard English?

Reviewer #1: No

Reviewer #2: Yes

5. Review Comments to the Author

Reviewer #1: Dear,

please completely describe about the variables and scales used for measurement the variables. name of scales and psychometric properties in your context. title also needs revision. name of variables needs to close each other and context should be at the end. year of study added. say more implications of study for your context in conclusion.

Good Luck

Reviewer #2: 1. Means and standard deviations should be reported together in accordance with PLOS One style.

2. Please remove the second column of Table 6. Repeating “Person Correlation” is unnecessary.

3. Please separate the discussion from the conclusion.

4. Please review and correct text for omission and other errors (for example, line 385 included p 0.05).

5. It is more reader-friendly to state the text of the hypothesis instead of displaying H1, H2, H3…

6. PLOS authors have the option to publish the peer review history of their article (what does this mean?). If published, this will include your full peer review and any attached files.

Reviewer #1: No

Reviewer #2: **Yes: **Dr. Ali M. AL-Asadi

---

## [Author Response · Author response to Decision Letter 0]

16 Jan 2024

Dear Bo Pu, Ph.D.,

We appreciate the thorough review conducted by the academic editors and reviewers on our manuscript titled "The Impact Environmental Education at Chinese Universities on College Students' Environmental Attitudes" submitted to PLOS ONE. We have carefully considered each comment and made the necessary revisions to address the concerns raised. Below, we respond to each point raised by the academic editors and reviewers:

Academic Editors' Comments:

1. File Naming and Style Requirements:

We have revised the file naming according to PLOS ONE's style requirements.

2. Raw Data Deposit:

We acknowledge the importance of data accessibility and have deposited our raw data in a repository to enhance visibility, appreciation, and citation. The data is stored in a GitHub repository, accessed at:

https://github.com/LeeYibo/Dissertation-data-on-ecological-attitudes

3. Title Correction:

We have changed the title of the article to "The Impact of Environmental Education at Chinese Universities on College Students' Environmental Attitudes" and ensured that the title is consistent between the online submission form and the manuscript. 

4. Reference List:

The reference list has been thoroughly reviewed and corrected. No retracted papers in the reference list.

Reviewer #1's Comments:

1. Description of Variables and Scales:

We have provided a comprehensive description of the variables and scales used for measurement, including their names and psychometric properties in our context.

2. Title Revision:

The title has been revised (Academic Editors' Comments: 3.)

3. Year of Study:

The year of study has been added to the manuscript for clarity. ( 3.2. data sources and basic information about the sample)

4. Implications in Conclusion:

We have expanded the conclusion section to include more explicit implications of the study for our context.

Reviewer #2's Comments:

1. Reporting Means and Standard Deviations:

Means and standard deviations have been reported together as per PLOS ONE style guidelines. (Table 4. &Table 5.)

2. Table 6 Modification:

The second column of Table 6, containing redundant information, has been removed for clarity. (remove the second column: Pearson's correlation)

3. Separation of Discussion and Conclusion:

We have separated the discussion and conclusion sections to enhance the structure of the manuscript.

4. Text Review and Correction:

We have thoroughly reviewed the manuscript and corrected any omissions and errors. The corrections are indicated in Revised Manuscript with Track Changes through the revision mode (Including the correction of line 385 regarding statistical significance, revised line 388.)

5. Hypothesis Presentation:

We have revised the presentation of hypotheses to state them in a more reader-friendly manner.

We believe that these revisions substantially improve the quality and compliance of our manuscript with PLOS ONE guidelines. We appreciate the time and effort invested by the academic editors and reviewers in evaluating our work.

Sincerely,

Yibo Li

PONE-D-23-29584

---

## [Decision Letter · Decision Letter 1]

6 Feb 2024

The Impact of Environmental Education at Chinese Universities on College Students' Environmental Attitudes

PONE-D-23-29584R1

Dear Dr. Li,

We’re pleased to inform you that your manuscript has been judged scientifically suitable for publication and will be formally accepted for publication once it meets all outstanding technical requirements.

Kind regards,

Bo Pu, Ph.D.

Academic Editor

PLOS ONE

Additional Editor Comments (optional):

this manuscript should be published.

Reviewers' comments:

Reviewer's Responses to Questions

**Comments to the Author**

1. If the authors have adequately addressed your comments raised in a previous round of review and you feel that this manuscript is now acceptable for publication, you may indicate that here to bypass the “Comments to the Author” section, enter your conflict of interest statement in the “Confidential to Editor” section, and submit your "Accept" recommendation.

Reviewer #2: All comments have been addressed

2. Is the manuscript technically sound, and do the data support the conclusions?

Reviewer #2: Yes

3. Has the statistical analysis been performed appropriately and rigorously? 

Reviewer #2: Yes

4. Have the authors made all data underlying the findings in their manuscript fully available?

Reviewer #2: Yes

5. Is the manuscript presented in an intelligible fashion and written in standard English?

Reviewer #2: Yes

6. Review Comments to the Author

Reviewer #2: (No Response)

7. PLOS authors have the option to publish the peer review history of their article (what does this mean?). If published, this will include your full peer review and any attached files.

Reviewer #2: **Yes: **Dr. Ali M. AL-Asadi

---

## [Editor Report · Acceptance letter]

20 Feb 2024

PONE-D-23-29584R1 

PLOS ONE

Dear Dr. Li, 

I'm pleased to inform you that your manuscript has been deemed suitable for publication in PLOS ONE. Congratulations! Your manuscript is now being handed over to our production team.

Kind regards, 

on behalf of

Dr. Bo Pu 

Academic Editor

PLOS ONE